# AxonNet: A Self-supervised Deep Neural Network for Intravoxel Structure Estimation from DW-MRI

**Hanna Ehrlich**[1]                                                                                       HANNA.EHRLICH@CIMAT.MX

**Mariano Rivera**[1]                                                                                    MRIVERA@CIMAT.MX

[1]*Centro de Investigacion en Matematicas AC, Guanajuato, Gto., 36000 Mexico.*

## Abstract

We present a method for estimating intravoxel parameters from a DW-MRI based on deep learning techniques. We show that neural networks (DNNs) have the potential to extract information from diffusion-weighted signals to reconstruct cerebral tracts. We present two DNN models: one that estimates the axonal structure in the form of a voxel and the other to calculate the structure of the central voxel using the voxel neighborhood. Our methods are based on a proposed parameter representation suitable for the problem. Since it is practically impossible to have real tagged data for any acquisition protocol, we used a self-supervised strategy. Experiments with synthetic data and real data show that our approach is competitive, and the computational times show that our approach is faster than the SOTA methods, even if training times are considered. This computational advantage increases if we consider the prediction of multiple images with the same acquisition protocol.

**Keywords:** Self-supervised neural network, Axonal structure estimation, DW-MRI, Multitensor.

## 1. Introduction

Axonal structure estimation from Diffusion Weighted MRI (DW–MRI) data consists on to estimate the preferred orientation of the water diffusion in brains which is usually constrained along the axon orientations. The analysis of DW-MRI allows one to estimate neural connectivity patterns in vivo (Daducci et al., 2014). Application of such structure and connectivity patterns are the study of sex differences (Ryman et al., 2014), brain structure discovery (Maller et al., 2019), neurological disorders (Maller et al., 2019) and brain deceases (Assaf and Pasternak, 2008; Rose et al., 2008), among many others. It has been remarked that to estimate connectivity pattern the recovered locally axonal structure needs to be reliable (Maller et al., 2019).

The Diffusion Tensor (DT) model is maybe the most popular one for explaining the diffusion MRI signal $S(\mathbf{g}_i, b_i)$ in a voxel with a unique axonal fiber. This model constructs on the Stejskal-Tanner equation (Basser et al., 1994; Basser, 1995):

$$S(\mathbf{g}_i, b_i) = S_0 \exp\left(-b_i \mathbf{g}_i^\top \mathbf{D}\, \mathbf{g}_i\right) \tag{1}$$

where $\mathbf{g}_i$ is a unit vector (associated with a magnetic gradient) defining the $i$–th acquisition direction, $b > 0$ is a predefined value representing parameters of acquisition, $\mathbf{D} \in \mathbb{R}^{3\times3}$ is the co-variance matrix of the diffusivity and $S_0$ is the measured signal with no diffusion weighting (the case when $b = 0$). Matrix $\mathbf{D}$ is a positive semi-definite (indeed, positive definite by physical reasons) squared matrix with the six free parameters. Its eigenvectors provides the model's orientation and correspond to the axes of the ellipsoid representing

the diffusion. Meanwhile, the eigenvalues provides diffusion magnitudes. In the case of a single axonal bundle, the eigenvalues satisfy: $\lambda_1 > \lambda_2 \approx \lambda_3$. The first eigenvector is the main diffusion tensor direction (fiber orientation), the second and third are orthogonal to this one. According with Jeurissen et al. (2013), crossing fibers are present in 60–90% of the diffusion data. Such voxels's signal can be represented with the Diffusion Multi-Tensor Model by a linear combination of $t$ single Diffusion Tensor Models, each one with its corresponding parameters. The Multi-tensor model is expressed as follows (Tournier et al., 2011; Ramirez-Manzanares et al., 2007):

$$S(\mathbf{g}_i, b_i) = S_0 \sum_{j=1}^{t} \alpha_j \exp\left(-b_i \mathbf{g}_i^\top \mathbf{D}_j \mathbf{g}_i\right) + \eta \tag{2}$$

$$\text{with} \quad \mathbf{1}^\top \alpha = 1 \tag{3}$$

$$\alpha > 0.1 \tag{4}$$

$$\alpha_j \leq \alpha_k, j < k \tag{5}$$

where, one has a matrix $\mathbf{D}_j$ for each tensor, the elements of $\alpha$ vector are the mixture coefficients (volume fractions), $\eta$ is the noise in the data, $\mathbf{1}$ is a vector with entries equal to one and which size depends on the context, constraint (4) is a form of non–negativity robust to noise and (5) imposes an artificial order useful in out notation.

We present a self-supervised strategy for analyzing DW-MRI: it consists of a proper data representation based on a formal generative method and a simple neural network for computing the fiber distribution. It is flexible to different acquisition protocols and can be adapted to process data in a voxelwise or patches-wise manner. It is computationally efficient and accurate. Recently there is attention for acquiring HARDI and super-HARDI DW-MR images (Maller et al., 2019) which analysis demands faster methods, as proposed.

## 2. Artificial Neural Network for DW-MRI Analysis

Analyzing DW–MRI data is equivalent to estimate the parameters for the model (2)–(5), or of any chosen generative model, given the acquired DW-MRI data. In recent years, Deep Learning models (DL) has been used to approach this task by the reconstruction of fiber Orientation Distribution Functions (ODF). It has been tackled as a classification approach (Koppers and Merhof, 2016) with a Convolutional Neural Network (CNN) or a regression approach with different architectures: 3D-CNN in (Lin et al., 2019), a Spherical U-Net in (Sedlar et al., 2020) and, a U-Net and a HighResNet in (Lucena et al., 2020); the first two allowing a signals neighborhood patch as input.

We assume that the lack of labeled real data (a golden standard) is a constant in this task: the development of reliable analysis methods for DW–MRI continues nowadays. Moreover, acquisition protocols may vary between acquisitions. In our work, we have chosen to develop a self–supervised scheme: we train Artificial Neural Networks (ANNs) with synthetic signals generated with direct models using the parameters which define our acquisition protocol. Then the trained models are used in real signal to infer the voxel axonal structure.

## 2.1. Self-supervised strategy

A DW-MR image $\mathcal{I} = \{\mathbf{S}\}$ is composed of spatially (3D) correlated signals. Each signal is assumed to be a measure of the Diffusion Multi-Tensor Model. The predefined acquisition protocol includes the set $\mathcal{G} = \{(\mathbf{g}_i, b_i)\}$. This contains $n$ pairs of gradients $\mathbf{g}_i$ and its correspondent $b_i$ values, where $\|\mathbf{g}_i\| = 1$, $i = 1, 2, ..., n$. A voxel's signal $\mathbf{S} \in \mathbb{R}^n$ is a vector with the entries $S(\mathbf{g}_i, b_i)$; see (2). We compute our datasets by generating independent synthetic signals of voxel's neighborhood (patch) of size $3 \times 3 \times 3$ for a preset acquisition protocol $\mathcal{G}$. We simplify our models by assuming normalized signals; ı.e, $S_0 = 1$. It can be achieved in real data by acquiring the $S_0$ signal (corresponding to $b = 0$) and normalizing the image's signals. We also assume equal 3 the maximum number of tensors.

Now, given a DW-MRI volume acquired with a set of parameters $\mathcal{G}$, we estimate the diffusion tensor of reference by focusing on the *corpus callosum* (a brain zone characterized by having a single coherent fiber population). For example, $\lambda = [0.0014, 0.00029, 0.00029]$ were the eigenvalues corresponding to the single tensor model of an analyzed DW–MRI volume. Thus, to construct a neighborhood sample, we randomly set the noise level with a SNR $\in [20, 30]$ simulating Rician noise; Rician distribution is generally assumed for noise in MRI data (Nowak, 1999). Then, a unique volume fractions $\alpha$ is randomly generated according with (2)–(5) and set equal for the voxels in the patch. In this manner, the signals in a patch share the volume fraction and noise level. Thus, we generate the angles $(\theta_2, \theta_3)$ from a uniform distribution in range $[0, \pi]$ rad. Hence, we set the relative Principal Diffusion Directions (PDDs) equal to $[1, 0, 0]^\top$, $[1, \cos\theta_2, 0]^\top$ and $[1, 0, \cos\theta_3]^\top$, for the first, second and third components; respectively. Next, the PDDs are randomly rotated with rotation angles uniformly distributed in the sphere. In this moment we have a set of PDDs, $\mathbf{d}$, following we explain how a PDDs set is assigned to each voxel in the patch and how slight orientation changes are introduced. We generate the PDDS for the voxels at the corners of the patch of $3 \times 3 \times 3$ voxels: $\mathbf{d}_k = \mathbf{d} + n_k$, for $k \in C$ the index set corresponding to the voxels at the corners, $n_{k,i} \sim \mathcal{N}(0, \sigma_r^2)$; we choose $\sigma_r = 0.14$ (approximately 8 degrees). The PDDs of reminder voxels in the patch are trilinear interpolated. Finally, we normalize each PDD. Once we defined the parameters for the patch, the signal of each voxel is generated according to model (2)–(5); this process can be expressed mathematically as $S = F(\alpha, \mathbf{d}; b, \mathbf{g})$. We use the Diffusion Multi-Tensor Model implementation in the DiPy library (Garyfallidis et al., 2014) to generate the signals. Figure 1 shows the Orientation Distribution Function (ODF) of a generated patch. One can appreciate smooth variations of the tensors orientation.

Our goal is to predict the parameter vectors $\mathbf{d}$ and $\alpha$ for each tensor $j$ composing a signal. This ill-conditioned inverse problem that can be expressed by

$$\alpha, \mathbf{d} = F^{-1}(S; b, \mathbf{g}) \tag{6}$$

when, in general, $F^{-1}$ does not exists. Hence, it is important to use a variable encoding that facilitates estimating from examples the relationship between parameters and signals. Previous works have used a set (dictionary) of fixed signals (atoms), then they compute a weighted combination of atoms that fit the original signal (Ramirez-Manzanares et al., 2007). We also build upon the dictionaries strategy in parameter encoding. We compute a dictionary $\mathcal{D} = \left[\tilde{\mathbf{d}}_k\right]$ with $m = 362$ PDDs uniformly distributed in the hemisphere. Then, for each voxel, we find the PDDs in dictionary nearest to the generated PDD: $k_j^* =$

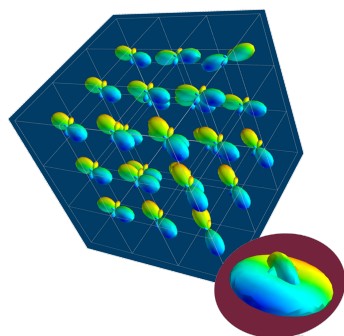

**Figure 1:** Neighborhood representation

$\text{argmin}_k |\tilde{\mathbf{d_k}}^\top \mathbf{d_j}|$. Hence, we set the coefficient dictionary $\tilde{\alpha}_{k_j^*} = \alpha_j$ for $j = 1, 2, 3$ and $\tilde{\alpha}_{l \notin \{k_j^*\}} = 0$. We design the ANNs models to estimate the $\tilde{\alpha}$ coefficients. However, because the dictionary is discrete, the dictionary's PDDs do not coincide with the generated signals' PDD. To reduce this inconvenient, we include a confusion matrix in comparison metrics, so that we compare the estimated $\tilde{\mathbf{d}}_{k_j^*}$, does not with the real $\mathbf{d}_j$, but with $\mathbf{W}^\sigma \mathbf{d}_j$; where $\mathbf{W}^\sigma$ is a matrix of Gaussian weights. That allows the model to better learn the labels without penalizing small orientation misalignment. $\mathbf{W}^\sigma \in \mathbb{R}^{m \times m}$ is a symmetric matrix where each row $\mathbf{W}_i^\sigma$ contains the blurring weights for the $i$-th dictionary direction, which are directed related to the angles between directions. Satisfying $\mathbf{W}_{ik} < \mathbf{W}_{ij} < 1$ if the $j$-th direction is nearest than $k$-th direction to the $i$-th one. Then small Gaussian Labels, $\mathbf{W}^\sigma \mathbf{d}_j < 1e - 3$ are clipped to zero values, normalized and used as the target for training the models.

## 2.2. Voxel Model

As mentioned before, each acquisition protocol needs a different analysis model because it may implies different input size and/or different acquisition parameters. It is not realistic to design a global model able to analyze any DW-MRI image. Our method can be separated in two steps. First, the training data generation that capture most of the acquisition protocol particularities, presented in previous subsection). Second, the analysis method based on a general and simple Neural Network (NN) model: multilayer perceptron (MLP). Since we implement an *ad hoc* data codification, the proposed neural network model is relatively simple: a MLPs. Consequently, our models are fast to train and fast for computing the predictions (the DW-MRI analysis). Simple models can be easily adapted to different acquisition protocols. NN models for different acquisition protocols have similar architectures: they only modify the input data size and some architectural hiperparameters.

Our MLP predicts the $\tilde{\alpha}$ coefficients given the signals in a voxel. The model's input is the signal $\mathbf{S}$ (flatten vector of size $n$) taken from the central voxel in the generated patch. The output is the $\tilde{\alpha}$ coefficients (flatten vector of size $m$) associated with PDDs in the dictionary $\mathcal{G}$. Figure 2 illustrates the voxelwise model.

We implement the Voxel model as an MLP with six dense layers. The first five layers have a ReLU activation function. The last one uses the sigmoid activation: the output is in range $[0, 1]$. We include a dropout layer, with a 0.2 rate, previous final dense layer to reduce the overfitting risk. We select Mean Squared Error (MSE) as a loss function for the

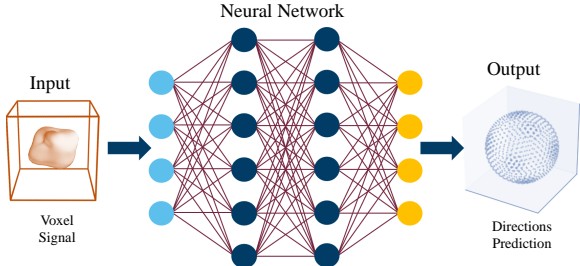

**Figure 2:** Voxel Model

training phase; we also investigate the mean absolute error without a clear advantage. We investigate several optimization algorithms, and we obtain the best results using ADAM algorithm (Kingma and Ba, 2014): (fast convergence, resiliency to overfitting, and accurate results). The learning rate was set equal to $1 \times 10^{-4}$. A learning rate decay equal to $1 \times 10^{-6}$ avoids stuck the training. Appendix A presents the MLP's architecture details for the Voxel model.

Once the voxel model is trained, the prediction on real data is straight forward, voxel to voxel. The output corresponds to the same coordinates than the predicted signal. Our implementation predicts the entire image (volume) allocating the signals in a single batch.

## 2.3. Neighborhood Model

Our Neighborhood model tries to incorporate spatial context of a voxel's signal. The signals from a voxel patch of size $3 \times 3 \times 3$ are taken, it give us a 4D input $(3, 3, 3, n)$ which is flatten to convert it into the input vector of size $27n$. The output remains without changes: the prediction for the patch's central voxel of the coefficients $\tilde{\alpha}$ associated with PDDs in a hemisphere. Figure 7 in Appendix A illustrates the model, compare with Figure 2.

The architecture of MLP that implements the neighborhood-wise model is similar to the voxel-wise model one. However, we increase the number of neurons in hidden layers to process additional information of adjacent voxels. The training process uses the same principle as the voxel model: MSE as loss function and ADAM as training algorithm. In this case, we reduce the learning rate to $1 \times 10^{-5}$. Despite we found that the learning rate decay has a more relevant function because it is more likely to be trapped by bad local minima, the value $1 \times 10^{-6}$ accomplishes the task. We introduce a zeros margin of size one (padding) to analyze an entire DW-MRI in the prediction stage. It increases by two the $x$, $y$, and $z$ dimensions. Appendix A presents the MLP's architecture details for the Neighborhood Model.

**Implementation detail**. We present our strategy for the simultaneous prediction of the entire volume instead of predicting voxel by voxel. The idea is to produce as many $3 \times 3 \times 3$ patches as voxels in the image, each patch center at each voxel. Then, we can generate a data batch that contains adjacent and non-overlapped patches with an adequated slicing-reshape strategy: if the coordinates $(i, j, k)$ correspond to a predicted central voxel for a given patch, then the voxels with coordinates $(i+3\ell_i, j+3\ell_j, k+3\ell_k)$ are also predicted in the

batch, providing the voxel is in the volume; where $\ell_i, \ell_j, \ell_k \in \mathbb{Z}$. Based on this observation, we note that the complete image can be processed with 27 batches. It is just necessary to slide the image in $3\times3\times3$ patches once and slip it twice by one element in each direction to have all the predictions. Figure 3 illustrates how the algorithm works. The first row depicts different patch batches (partitions). The blue diagram represents the partition to predict the front upper right corner. Yellow, green, and orange diagrams illustrate the resultant partitions of slipping the original one by one element: to the left, to the bottom, and to back; respectively. Three dots represent the remainder partitions needed to process the full image. The second row illustrates the voxels predicted corresponding to each partition. In the third row is depicted the reconstruction of the entire image, the final result. Our implementation takes care of the problems that can occur if it is not paid attention to the image size.

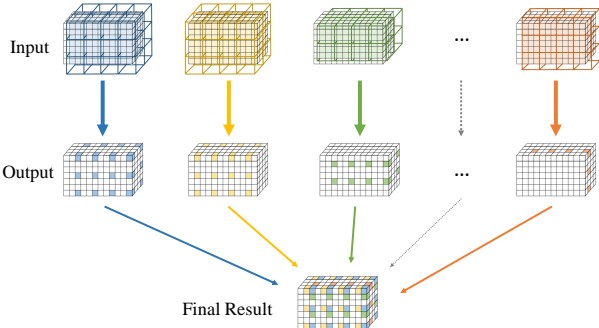

**Figure 3:** Prediction by slices.

## 3. Results

Voxel and Neighborhood models were tuned and trained using the synthetic data for each dataset generated with the same gradient table as the original. The performance of both models was tested over the test synthetic dataset, obtaining qualitatively and quantitative results.

**Training stage**. We compare the results of the proposed Voxel (VOX) and Neighborhood (NBH) models with two of the SOTA methods which tackle the same task: Diffusion Basis Functions (NNLS) proposed by Ramirez-Manzanares et al. (2007)) and Constrained Spherical Deconvolution (CSD) proposed by Tournier et al. (2007). There are many options to compare distributions, a common comparisson procedure used in this context is to detect peaks and compute the angular error between the real peaks and the estimated ones. However, to compare modes in not a standard procedure for comparing distributions. Among them two notable options are Kullback-Leibler (KL) Divergence and the Wasserstein Distance (also know as the Earth Mover Distance, EMD). Despite its computational cost, EMD has shown to represents more precisely the distribution distance (Levina and Bickel, 2001; Aranda et al., 2011; Arjovsky et al., 2017). EMD represents the minimum cost of transforming a peak distribution into another, weighting by angle. We create a synthetic dataset with gradient table of the Stanford HARDI dataset Rokem et al. (2015), the eigenvalues of a Diffusion Tensor model fitted to the corpus callosum region, and the SNR computed in such a data (Descoteaux et al., 2011). The estimated SNR depends on

image region: most of the measures laid into $[20, 24]$, so we randomly generate data selecting the SNR into $[20, 30]$. Figure 4 depicts the error for each analyzed model. The vertical axis corresponds to the angle ($\theta_1$) between the first PDD and the second one. Meanwhile, the horizontal axis shows the angle between the third PDD and the plane formed by the first two PDDs. The dynamic range of the error maps shows a better performance of the proposed models. We select some predictions for a visual inspection (qualitatively comparison). For illustration purposes, we choose one between the top–10 and one of the bottom–10 according to its EMD values for the studied models: VOX, NBH, NNLS, and CSD. The results are presented in Figure 5. The first two columns correspond to the best predictions: the first column shows the target and the second column shows the prediction. The third and fourth columns follow the same order but for the worst predictions. Arrows illustrate the generated PDDs (ground truth). According to the $\alpha$ value: blue, orange, and green were used for the first, second, and third PDD, respectively.

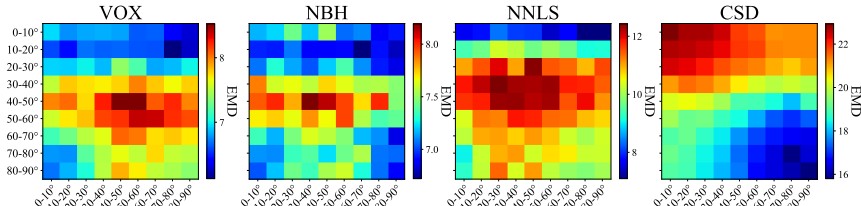

**Figure 4:** EMD (error) heat-maps by model predictions

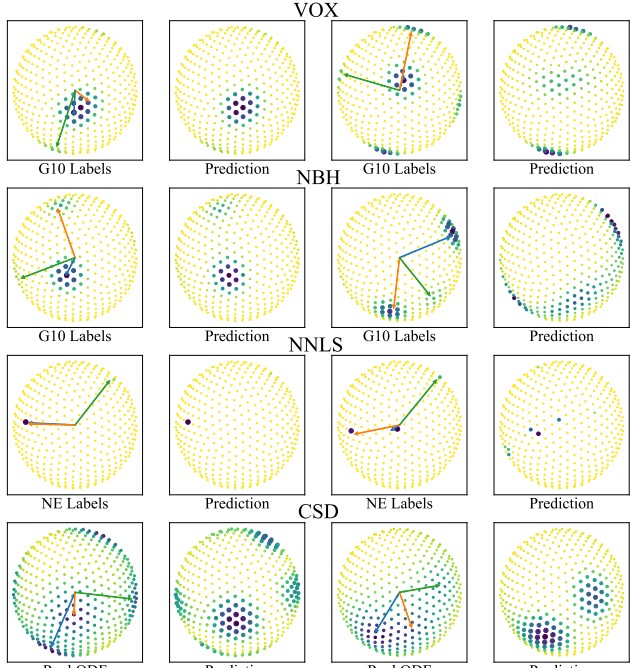

**Figure 5:** Predicted voxels with lowest (first two columns) and highest error (last two columns).

**Table 1:** Computational time for prediction, in minutes.

| Dataset | VOX | NBH | CSD | NNLS |
|---|---|---|---|---|
| **Stanford** | 0.23 | 0.81 | 6.08 | 27.08 |
| **Local** | 0.26 | 1.90 | 8.74 | 114.32 |

**Inference stage**. Now we present results of the inference stage with two DW-MRI real datasets. First, the free access dataset Stanford HARDI Rokem et al. (2015) included in DIPY Library, with dimension (81,106,76) voxels and 160 signals per voxel (number of gradients). The acquisition protocol composed uses 150 gradients with b-value equal 2000 and 10 with b-value equal zero. Second, a local DW-MRI with $(128, 128, 70)$ voxels with 64 gradients with b-value 1000 plus 1 gradient with b-value 0, each of them is repeated 5 times resulting in signals of size 325. Training time for our models by depends on datasets: The Voxel model takes $1.24sec.$ for the Stanford HARDI and $1.24sec.$ for out local dataset. Meanwhile, Neighborhood model takes $1.91sec.$ for the Stanford HARDI and $3.34sec.$ for our local dataset. Prediction times are shown in Table 1. Figure 6 compares the final results in a Stanford dataset slice, showing the local detected structure with the studied models. More slices results are presented in Appendix C.

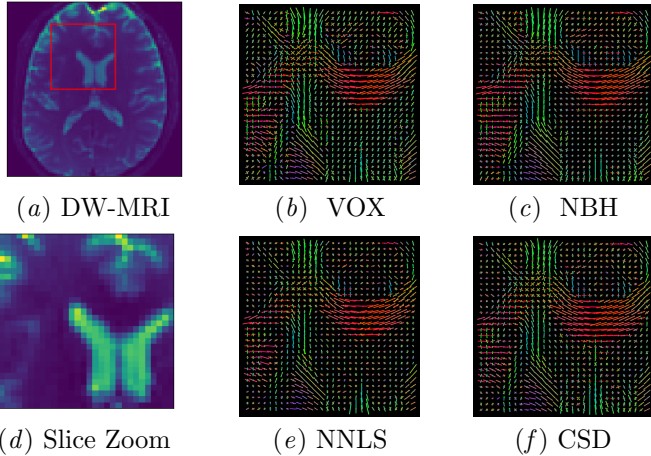

(*a*) DW-MRI     (*b*) VOX     (*c*) NBH

(*d*) Slice Zoom     (*e*) NNLS     (*f*) CSD

**Figure 6:** Predicted intravoxel structure in real data with the compared models.

## 4. Conclusions

We propose a strategy for analyzing DW-MRI based on two stages: a proper data representation based on a formal generative method (Diffusion Multitensor) and a simple neural network (MLP) for computing the fiber distribution. Labeled data are unneeded in our self-supervised strategy. We only require an estimate of diffusion parameters and the gradients of the acquisition protocol. We demonstrate by experiments that our approach is flexible by generating solutions for two datasets with different acquisition protocols and two kinds of support regions (voxelwise and patches-wise). Also, it compares favorably with SOTA methods: it processes entire volumes in a fraction of the time than SOTA methods and with comparable (or even better) error metrics.

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

# Appendix A. Architecture detail of the MLP models

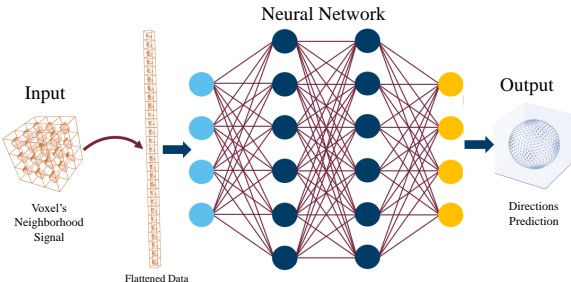

**Figure 7:** Neighborhood Model illustration. The patch of signals is flatten to be the input, the output (labels) keeps the same the size as in the Voxel model.

Next figures describe the details of MLPs' models. The dense layers include bias and the dropout rate was set equal to 0.2

**Model: "VoxelModel"**

| Layer Type | Activation Function | Input Shape | Output Shape | Param # |
|---|---|---|---|---|
| Dense_1 | ReLU | (None, 160) | (None, 2048) | 329728 |
| Dense_2 | ReLU | (None, 2048) | (None, 1024) | 2098176 |
| Dense_3 | ReLU | (None, 1024) | (None, 1024) | 1049600 |
| Dense_4 | ReLU | (None, 1024) | (None, 512) | 524800 |
| Dense_5 | ReLU | (None, 512) | (None, 512) | 262656 |
| Dropout | - | (None, 512) | (None, 512) | 0 |
| Output (Dense) | tanh | (None, 512) | (None, 362) | 185706 |

Total params: 4,450,666
Trainable params: 4,450,666
Non-trainable params: 0

Loss function : mean_squared_error
Optimizer : Adam
Learning Rate : 0.0001
Decay : 1e-06

**Figure 8:** Voxel Model Architecture.

**Model: "NeighborhoodModel"**

| Layer Type | Activation Function | Input Shape | Output Shape | Param # |
|---|---|---|---|---|
| Dense_1 | ReLU | (None, 4320) | (None, 4096) | 17698816 |
| Dense_2 | ReLU | (None, 4096) | (None, 2048) | 8390656 |
| Dense_3 | ReLU | (None, 2048) | (None, 1024) | 2098176 |
| Dense_4 | ReLU | (None, 1024) | (None, 1024) | 1049600 |
| Dense_5 | ReLU | (None, 1024) | (None, 512) | 524800 |
| Dense_6 | ReLU | (None, 512) | (None,512) | 262656 |
| Dropout | - | (None, 512) | (None, 512) | 0 |
| Output (Dense) | tanh | (None, 512) | (None, 362) | 185706 |

Total params: 30,210,410
Trainable params: 30,210,410
Non-trainable params: 0

Loss function : mean_squared_error
Optimizer : Adam
Learning Rate : 0.0001
Decay : 1e-06

**Figure 9:** Neighborhood Model Architecture.

## Appendix B. Selection of de MLP hyperparameters

The appendix shows conducted some experiment to select hyper-parameters of our methods.

We investigate MSE and MAE metrics, the combinations were repeated 5 times each one and we observed if some of them appears to be unstable. For Voxel Models, all the repeated combinations show similar performance between them, but some repeated Neighborhood Models got stuck on a plateau, such corresponding combinations were also dismissed. We found that models using MSE as loss function lead to better results, specially if the hyperbolic tangent (tanh) is used as activation function for the last layer. In Figure 10 we can see the evolution of the loss function evaluated in the validation set for four Voxel Models. The models were trained using MSE as loss function and a learning rate of $1e-4$ using ADAM and RMSprop as optimization algorithms and sigmoid and tanh as activation functions for the last layer; RMSprop is unpublished, G. Hinton in Lecture 6e, Coursera Class.

Figure 11 shows the loss evolution for the Voxel Model (learning rate $1e-5$ have a slower MSE loss convergence). The experiments indicate that tanh is best choice as activation function that the sigmoid and that ADAM algorithm has a smoother convergence that RMSprop algorithm.

Figure 12 shows the mean plot of 5 repetitions by the 100 epochs for Stanford-HARDI and and Local datasets: neighborhood model learns better to predict the labels.

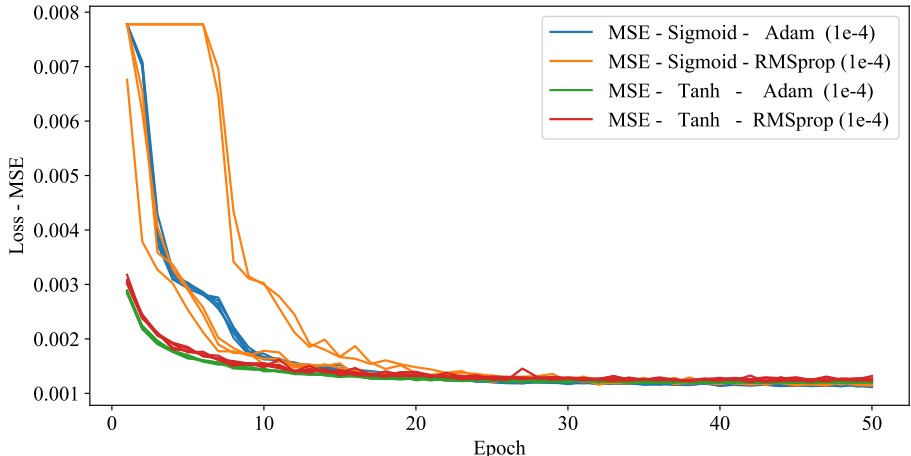

**Figure 10:** Voxel Models Training Process

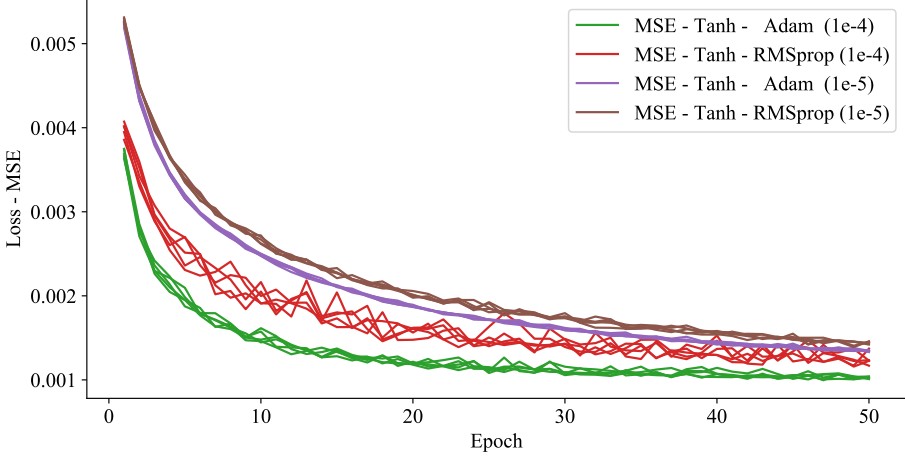

**Figure 11:** Loss value evolution during the training for Neighborhood Models.

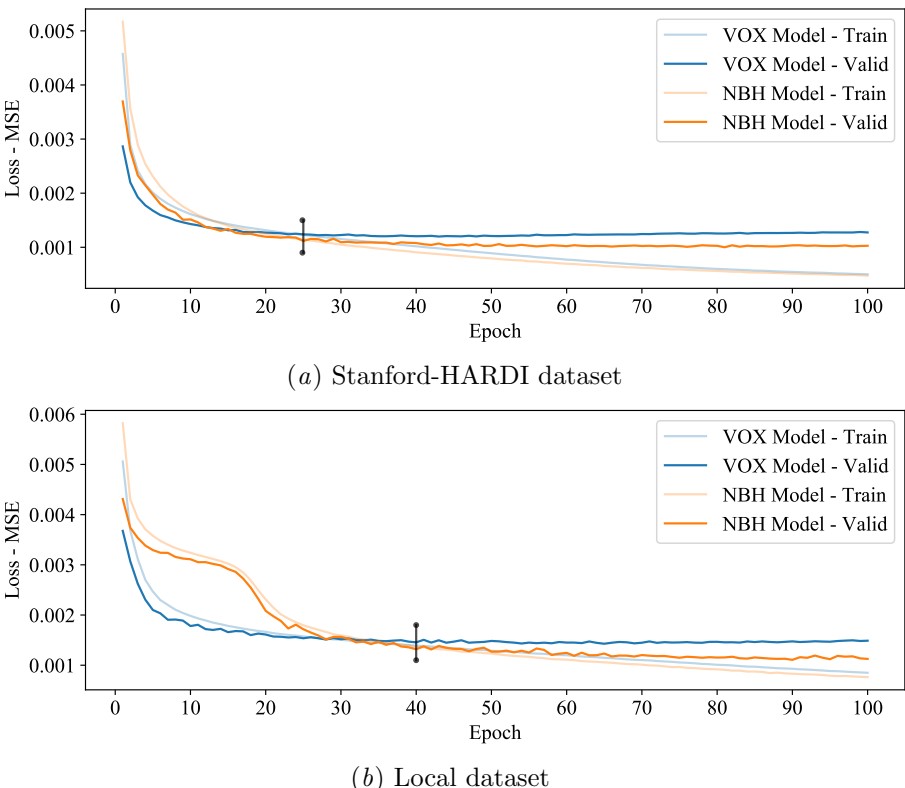

(*a*) Stanford-HARDI dataset

(*b*) Local dataset

**Figure 12:** Comparison of the loss value evolution during the training process for Voxel and Neighborhood Models.

## Appendix C. Axonal structure estimated in real data

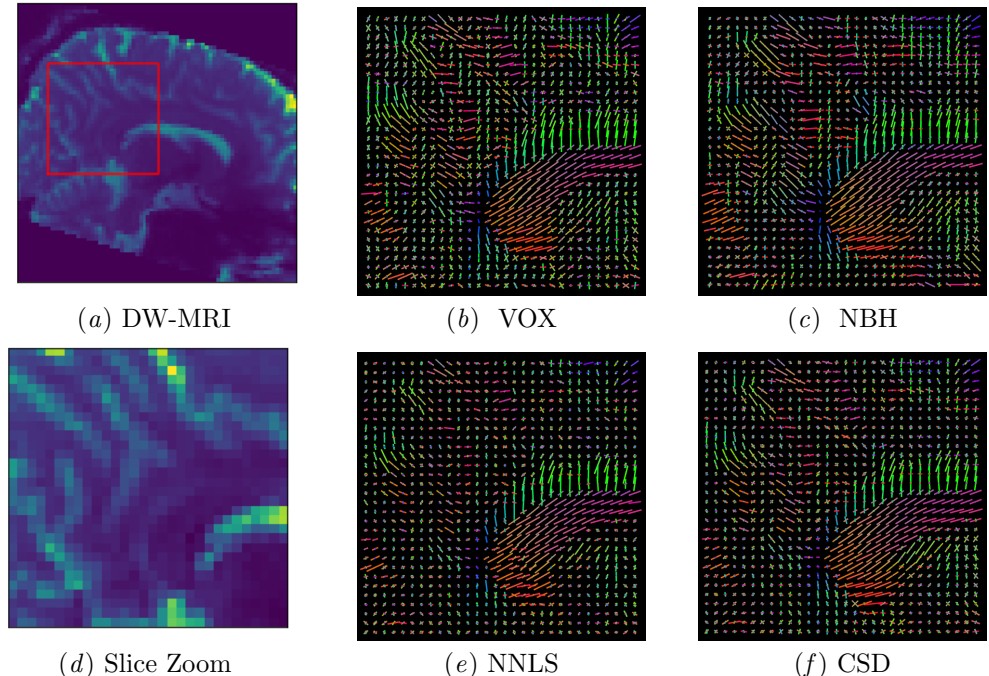

(a) DW-MRI      (b) VOX      (c) NBH

(d) Slice Zoom      (e) NNLS      (f) CSD

**Figure 13:** Example 2. Predicted intravoxel structure in real data.

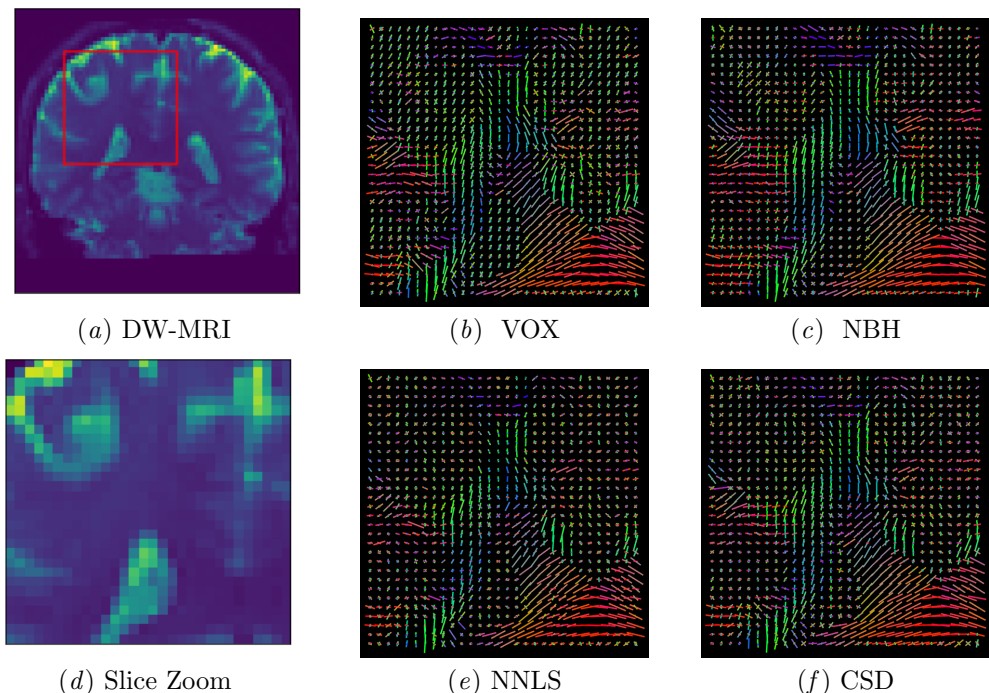

(a) DW-MRI      (b) VOX      (c) NBH

(d) Slice Zoom      (e) NNLS      (f) CSD

**Figure 14:** Example 3. Predicted intravoxel structure in real data.

