# OpenReview forum: "AxonNet: A Self-supervised Deep Neural Network for Intravoxel Structure Estimation from DW-MRI"
_MIDL.io/2021/Conference — Submitted to MIDL 2021_

### Official Review · AnonReviewer2 · 2021-03-06

**Confidence:** 5
**Preliminary Rating:** 1
**Final Rating:** 1

**Summary:**

The paper proposes an approach to the problem of estimating local fiber orientations in Diffusion Weighted Imaging data, which is a central problem in diffusion imaging. Authors propose a dense deep neural network in a self-supervised setting to estimate such orientations. Authors show qualitative results of their method on a synthetic dataset and the in-vivo human brain Stanford HARDI dataset (Rokem et al. 2015).

**Strengths:**

The paper proposes to use a self-supervised approach to estimate the local fiber orientations in diffusion data, which seems an interesting way to approach the problem. Following from their approach, the required or proposed neural network seems also appealing by its simplicity.

**Weaknesses:**

The methods section, although detailed, does not allow to have a clear understanding of what the input to the DL method is, or how the self-supervision signal is constructed. The experimental setting of the work is unclear, with misleading explanations on the data used and the experiments performed on each dataset. Also, the evidence provided does not seem sufficient to evaluate qualitatively the ability of the method to predict fiber orientations (only a heatmap of the prediction errors and qualitative figures are included). The relevant literature using DL methods for the presented problem is missing.

**Deanonymize Review:**

no

**Detailed Comments:**

- "Axonal structure" or "intravoxel parameters" are terms that are used inaccurately or too broadly when referring to what the presented DNN predicts, the intravoxel fiber orientation. Note that the fiber orientations do not correspond to an axonal structure, and are only descriptive of the orientational preference of coherent fiber groups through the measured diffusion signal, and is far from the resolution of an axon. Although the first sentence of the Introduction provides a somehow more complete explanation, it still incurs in the imprecision of stating that these are axon orientations; "axonal fiber bundle", used in the text, is a more appropriate term.
- "Application of such structure and connectivity patterns are the study of (...) brain deceases": the term "decease" is not appropriate here: it should be "disease".
- The diffusion tensor model was proposed in a work by Peter Basser dating back to 1994, so 8 years prior to the provided Basser and Jones (2002) reference on p.1.
- "(...) and λ1 is greater by approximately a 5 : 1 ratio". A reference would be necessary to support the statement.
- "Since it can be very hard to label (provide the ground truth of) real data, given that the acquisition protocol may vary between acquisitions.". The reason why providing a "ground truth" for real data is difficult is not the difference in acquisition protocols. Please provide the real underlying reason for this. In fact, authors do not perform experiments with different DWI acquisition protocols.
- The set S (in bold faces) should be defined the moment it is mentioned in Section 2.1.
- "In real data a pre-processing has to be applied by acquiring S0 for each voxel through the system". This is inaccurate or should be re-worded: S0 is a single 3D volumetric image, having a possibly different value for each voxel, but is acquired at once, as any other regular diffusion image.
- Can authors provide a reference as to why the noise distribution used is Rician?
- F in equation 4 is not defined.
- Paragraph two in Section 2.2. contains a series of nonspecific statements, e.g. "a dense neural network adapted to the input we have and the output we expect to predict.". Is any DNN not adapted to its input data or the target it aims to predict?
- The paragraph "Once the voxel model is trained, the prediction on real data is straight forward, voxel to voxel. The output corresponds to the same coordinates than the predicted signal. Our implementation predicts the entire image allocating all the data signals in a single batch" seems not to provide any relevant information, especially in the last sentence.
- "The architecture of the neighborhood–wise model is practically the same than the developed for the voxel–wise model, just making bigger the hidden layers to capture the additional information of the adjacent voxels." Using "practically" is noninformative. How "bigger" (using the same term the authors use) is the "Neighborhood Model" network or are its layers compared to the "Voxel Model"? Are there any other changes to the network?
- It is unclear the relevance of paragraph "Instead of predicting all the image voxel by voxel, we focus on simultaneous predictions. (...)" on p. 6, or it should be better explained. A better or clearer depiction in figure 4 should also help. Using a 3x3x3 neighborhood and sliding that over the entire 3D volume to feed the neural network does not seem to need any further explanation, unless the input is still too large to be fed to the network and a further split is required.
- Note that the symbol "Z" on p. 6 is not defined.
- Figure 5 should be made larger.
- Figure 6 labels are not explained: what do "G10 Labels", "NE labels", "Real ODF" mean? The colorbar in Figure 6 is missing.
- Training times seem not to correspond to what it would be expected for a DNN; even assuming large computational resources, the presented training times seem too fast: "Training time for our models by depends on datasets: The Voxel model takes 1.24sec. for the Stanford HARDI and 1.24sec. for out local dataset. Meanwhile, Neighborhood model takes 1.91sec. for the Stanford HARDI and 3.34sec. for our local dataset.". The hardware used is not mentioned.. Also, note that the abbreviation for the unit "seconds" is "s" and not "sec".

**Final Rating Justification:**

The revised manuscript addresses some of the concerns raised in the first submission and shows further details (e.g. model architecture, justification for some choices), some fundamental concerns remain:
- The self-supervision strategy is still unclear; no method has access to ground-truth on in vivo data (all orientation prediction DL methods rely on estimations provided by conventional methods).
- It is still difficult as a credible comparison that the orientation predictions of a multi-tensor-based model beats a model providing fODFs. A conclusive comparison would have been against competing DL methods (only cited).
- Figures 4 and 5 (paragraph "Training stage") seem to be reported on the training data; the qualitative close-up figures on test set are not concluding.
- More common error measures (other than EMD) would show more clearly the benefits.
- A non-negligible number of grammar mistakes still affect the work.

The manuscript requires further work so as to be presented.


**Justification Of The Preliminary Rating:**

The rationale behind the proposed method is unclear, the methods section is confusing, and the data used is not described clearly. There is no real quantitative evidence provided to measure the accuracy or improvement of the method over the rest of the compared methods. There is virtually no discussion on the results, no limitation is mentioned, and the conclusions are weak. There is no mention (nor comparison) to other DL methods that have been proposed to estimate fiber orientations from DWI data.

**Paper Type:**

methodological development

**Questions To Address In The Rebuttal:**

- Section 2.1 is unclear: it is unclear how the self-supervision acts, and it is unclear what information is born by the "Gaussian Labels". It is unclear why a blurring is necessary. Aren't the 362 directions or atoms enough to estimate the fiber orientation with sufficient accuracy?
- The use of the term "self-supervision" might require further explanation: it is not clear what the input to the DNNs are, and how the supervision guides the learning.
- It is unclear why an MSE loss is used if the output of the network is a "(...) (coefficient) of each direction at the dictionary".
- How does the method perform on voxels containing crossing fiber populations? Is this evaluated? The Stanford dataset used to obtain the results in Figure 5 is a whole brain dataset. Do these results refer to orientations in all voxels?
- EMD is not a commonly used measure to quantify the prediction accuracy for fiber orientations. It also makes it harder to estimate how well the method compares to other methods proposed in literature.
- The construction of the synthetic dataset is not well explained and should be done prior to presenting the results. Using a HARDI-like gradient table for a multi-tensor model seems to provide an excessively high angular resolution and does not seem the most reasonable choice. Can the reasons for this choice be explained? Is this the same as using the Stanford HARDI dataset mentioned on p. 7? This is made even more confusing by the passage "Training time for our models by depends on datasets: The Voxel model takes 1.24sec. for the Stanford HARDI and 1.24sec. for out local dataset. Meanwhile, Neighborhood model takes 1.91sec. for the Stanford HARDI and 3.34sec. for our local dataset."
- Figures 7, 8, and 9 do not provide sufficient close-ups to qualitatively evaluate the improvement of the estimated orientations over its competitors.
- No relevant comparison to other works estimating fiber orientations from DWI data (e.g. Koppes et al. "Direct Estimation of Fiber Orientations using Deep Learning in Diffusion Imaging" MICCAI 2016;  Lin et al. "Fast Learning of Fiber Orientation Distribution Function for MR Tractography Using Convolutional Neural Network" Med Phys 2019; Sedlar et al. "Diffusion MRI fiber orientation distribution function estimation using voxel-wise spherical U-net" MICCAI 2020; Lucena et al. "Using convolution neural networks to learn enhanced fiber orientation distribution models from commercially available diffusion magnetic resonance imaging" arXiv 2020).
- The manuscript contains a non-negligible number of English grammar mistakes.

**Special Issue:**

no

---

### Official Review · AnonReviewer1 · 2021-03-08

**Confidence:** 5
**Preliminary Rating:** 1
**Final Rating:** 1

**Summary:**

The authors propose to learn the parameters of a 3-tensor model directly from diffusion MRI images, either on a per-voxel basis or from a 3x3x3 neighborhood. To do so, the authors use a fully connected neural network trained on a synthetic dataset generated from structured noise. The authors use a dictionary of possible directions as labels for the parameters of the multi-tensor model, formulating the context as a classification problem. The authors test their method on a synthetic dataset akin to the one used for training as well as a two real life subjects. Comparison is provided against two classical reconstruction methods.

**Strengths:**

The idea of the proposed method is certainly interesting: current classical dMRI signal modeling methods are computationally heavy and artificial neural networks have the advantage of being computationally efficient once trained.

**Weaknesses:**

Unfortunately, the article suffers from lack of motivation behind the proposed work, lack of novelty, as well as poor writing and missing results. In its current state, the article cannot be said to be ready for publication.

The method is described as self-supervised. However, it is unclear how that claim is backed up: the method is trained on an in-house synthetic dataset generated with structure noise informed by biological observations (a single fiber population tensor model obtained in the corpus callosum of a single subject). While the training procedure is not dependent on real life data, direct labels are still computed from a dataset. A self-supervised learning method would require no labels and instead be trained on a pre-text task, for example reconstructing the dMRI signal itself, and then moving on to the "main" task. Even then, the second experiment includes training on real data, with no information on how the labels were computed.

Moreover, ODFs are referred to throughout the text (notably in figure 1 and 6, and implied in figures 2 and 3). However, the article clearly states
> In our method, the most interest parameters to estimate, from the DW-MRI signal, are the tensors directions $\textbf{d}$, defined by the angles directions $\theta$, and the set of the volume fractions $\alpha$.

and

> The general objective of this work is to predict the parameter vectors $\textbf{d}$ and $\alpha$ for each tensor $j$ forming a signal.

It reads as the authors conflate multi-tensor modelling and orientation diffusion functions. However, these two methods are wildly different, require different computation and make different assumptions about the underlying signal. The two methods should not be rendered equivalent.

As mentioned, the reason for using neural networks to model the diffusion signal should be better motivated. Section 2 includes:

> Since it can be very hard to label (provide the ground truth of) real data, given that the acquisition protocol may vary between acquisitions.

The authors fail to mention the most important cause of the lack of ground-truth dMRI orientation data: dMRI is a non-invasive method and is most used *in-vivo*. Therefore, the real orientation of the anatomical structures cannot be directly accessed.

 The authors also did not mention prior work which tackles the same problem. A quick search can reveal several ([1,2], for example) similar methods which should have been cited and possibly used as comparison in the experiments.

Analogously, the experiments are ill-defined and fail to convey motivation for the proposed work. Since the authors generated the dataset, they have access to the ground truth orientation and could have measured the mean angular error between their reconstruction and the ground-truth. As it is presented, Figure 5 does not offer meaningful information on the reconstruction capabilities of the method.

Finally, the article suffers from a lack of structure and poor writing making reading the article unnecessarily confusing. The text should be as concise as possible, sections should be self-contained, bring to the reader all the necessary information without repeating anything. Concepts should be clearly explained. There are also tonal shifts to a more familiar tone throughout the article, which should be avoided in a scientific paper.

[1]: Tian, Q., Bilgic, B., Fan, Q., Liao, C., Ngamsombat, C., Hu, Y., Witzel, T., Setsompop, K., Polimeni, J. R., & Huang, S. Y. (2020). DeepDTI: High-fidelity six-direction diffusion tensor imaging using deep learning. NeuroImage, 219, 117017. https://doi.org/10.1016/j.neuroimage.2020.117017
[2]: Koppers, S., & Merhof, D. (2016). Direct Estimation of Fiber Orientations Using Deep Learning in Diffusion Imaging. In Machine Learning in Medical Imaging (pp. 53–60). Springer International Publishing. https://doi.org/10.1007/978-3-319-47157-0_7

**Deanonymize Review:**

no

**Detailed Comments:**

The article makes several claims that should be accompanied by citations. Some examples include

> The Diffusion Tensor (DT) model, proposed by Basser and Jones (2002), is maybe the most popular one for explaining [...]

> [...] is greater by approximately a 5 : 1 ratio [...]

>  [...], we focus in the corpus callosum, a brain zone characterized by having a single coherent fiber populations

> Previous works, with same aim than us, use [...]

The tone of the article should be kept as formal and neutral as possible.

> Now we have a matrix $\textbf{D}_j$ for each tensor

> The output is deployed in an hemisphere, although here is shown as an sphere, this is because symmetry is assumed and it is easier to understand the figure in that way.

Some parts could easily be omitted. For example,

> The task of analyzing DW-MRI data is to estimate the model (2) parameters, or of any chosen generative model~, given the acquired DW-MRI data~.

The last part of the sentence duplicates the beginning of the sentence.

> The models implemented to achieve the aim of this work are relative simple. They consist on a dense neural network adapted to the input we have and the output we expect to predict. ~As mentioned before, each acquisition protocol needs a different model because it implies a different input size. It is difficult to create a global model able to analyze (predict the parameter of) any DW-MRI image. However, to create a model per acquisition protocol seems to be a good approach. Despite this fact, models for different acquisition protocols does not differ its structure in a significant way, it affects basically two things: the input size and the training process~

No need to describe the varying size of volumes. Also, this problem could be solved by resizing and cropping volumes to a reference shape.

> ~Instead of predicting all the image voxel by voxel, we focus on simultaneous predictions. If the complete volume is divided in adjacent [...] In the third row a reconstruction of the entire image is represented as final result. Our implementation take care issues that could be presented if no special attention is put on the image size.~

No need to explain the batching abilities of neural networks.

While some parts are superfluous, others are missing details. For example, the definition of the architecture is missing layer size, and the training regime (how many epochs, how was the dataset split, etc.).

> The architecture of the neighborhood–wise model is practically the same than the developed for the voxel–wise model, just making bigger the hidden layers to capture the additional information of the adjacent voxels.

Adjectives such as "bigger" should be avoided and should be replaced by the actual value of the concept described.

> The training process uses the same principle than the voxel model: loss function is MSE and the [...]

The voxel model should have described these implementation choices before. Also, how come MSE is used as a loss ? Isn't this a classification problem ? Why not use cross-entropy ?

Figures throughout the text should have meaningful and complete captions and covey their purpose on their own.

**Final Rating Justification:**

No rebuttal has been provided by the authors nor have any issues raised with the proposed work been addressed. In its current state, the paper is not in a suitable form for publication.

**Justification Of The Preliminary Rating:**

This paper reads like a preliminary version and is definitely not in a position to be published. Extensive rewriting is necessary to address the comments made in the previous sections, notably on the lack of results, motivations, citations, etc.

**Paper Type:**

both

**Questions To Address In The Rebuttal:**

The authors must address the comments above.

**Special Issue:**

no

---

### Official Review · AnonReviewer3 · 2021-03-09

**Confidence:** 2
**Preliminary Rating:** 2

**Summary:**

The authors present a method for estimating of intravoxel parameters from a DW-MRI based on deep learning techniques. The authors show that deep neural networks (DNNs) have the potential to extract information from diffusion-weighted signals to reconstruct cerebral tracts. The authors present two DNN models: one that estimates the axonal structure in the form of a voxel and the other to calculate the structure of the central voxel using the voxel neighborhood.

**Strengths:**

1. The results shows the models ordered by prediction times are: VOX, NBH, CSD and NNLS. The differences between models are very significant.
2. The reported results demonstrated the proposed models are competitive or overcome compared methods.
3. Recently acquired HARDI and super-HARDI DW-MR images may require the analysis of such data demands faster methods, as the proposed ones.

**Weaknesses:**

The proposed methods are based on a proposed parameter representation suitable for the problem. Since it is practically impossible to have real tagged data for any acquisition protocol, the authors used a self-supervised strategy. The techinical novelty seems limited.

**Deanonymize Review:**

no

**Justification Of The Preliminary Rating:**

The paper is well-written and easy to follow. The reported results demonstrated the proposed models are competitive or overcome compared methods. However, comparison study is somehow limited. Conclusion is not yet convincing. Not sure if the authors can address these issues.

**Paper Type:**

both

**Special Issue:**

no

---

### Official Review · AnonReviewer4 · 2021-03-09

**Confidence:** 4
**Preliminary Rating:** 1
**Final Rating:** 1

**Summary:**

The paper presents a deep neural network based method to reconstructe Diffusion Tensor Image (DTI) from Diffusion Weighted Magnetic Resonance Images (DW-MRIs) in a self-supervised manner. The model predicts the parameter vectors d (tensor directions) and volume fractions alpha for Diffusion tensor based on the DWIs.

**Strengths:**

The paper tries to address an important inverse problem of estimating Diffusion Tensor Image from Diffusion Weighted Magnetic Resonance Images. The deep neural network based method could provide solutions that are faster than traditional approaches.

**Weaknesses:**

The paper is very difficult to read, and lacks coherence in addition to several typos throughout the paper.
The introduction talks only about theoretical background of reconstructing Diffusion Tensor Image from DWIs, but does not properly state the problem paper wants to solve, and does not position itself compared to other related works.

**Deanonymize Review:**

no

**Final Rating Justification:**

While the paper has been updated improving slightly the readability over the initial version, the issues other than the writing raised in the review are not yet addressed and no rebuttal is provided.

**Justification Of The Preliminary Rating:**

With the current state of the paper, it seems that it will be too difficult to address all the concerns in the rebuttal. The exact contribution of the paper, its positioning with respect to current literature, and the problem formulation are not clear.

**Paper Type:**

methodological development

**Special Issue:**

no

---

### Meta-Review · Area_Chair1 · 2021-03-29

**Recommendation:** Reject

**Metareview:**

All reviewers have brought up serious concerns and major issues of the submitted work. They all agreed that the current quality of the manuscript will need significant improvement, in terms of writing, model design, and quantitive analysis of experimental results, to publish at MIDL. Details are included in each reviewer’s feedback.

**Paper Type:**

methodological development

---

### Decision · Program_Chairs · 2021-03-31

Reject